# Uncomputably complex renormalisation group flows

James D. Watson [1] ✉, Emilio Onorati[1] & Toby S. Cubitt [1]

Renormalisation group methods are among the most important techniques for analysing the physics of many-body systems: by iterating a renormalisation group map, which coarse-grains the description of a system and generates a flow in the parameter space, physical properties of interest can be extracted. However, recent work has shown that important physical features, such as the spectral gap and phase diagram, may be impossible to determine, even in principle. Following these insights, we construct a rigorous renormalisation group map for the original undecidable many-body system that appeared in the literature, which reveals a renormalisation group flow so complex that it cannot be predicted. We prove that each step of this map is computable, and that it converges to the correct fixed points, yet the resulting flow is uncomputable. This extreme form of unpredictability for renormalisation group flows had not been shown before and goes beyond the chaotic behaviour seen previously.

Understanding collective properties and phases of many-body systems from an underlying model of the interactions between their constituent parts remains one of the major research areas in physics, from high-energy physics to condensed matter. Many powerful techniques have been developed to tackle this problem. One of the most far-reaching was the development by Wilson[1,2] of *renormalisation group* (RG) techniques, building on early work by others[3,4]. At a conceptual level, an RG analysis involves constructing an RG map that takes as input a description of the many-body system (e.g., a Hamiltonian, or an action, or a partition function, etc.), and outputs a description of a new many-body system (a new Hamiltonian, or action, or partition function, etc.), that can be understood as a "coarse-grained" version of the original system, in such a way that physical properties of interest are preserved but irrelevant details are discarded.

For example, the RG map may "integrate out" the microscopic details of the interactions between the constituent particles described by the full Hamiltonian of the system. This procedure generates a coarse-grained Hamiltonian that still retains the same physics at larger length scales[5]. By repeatedly applying the RG map, the original Hamiltonian is transformed into successively simpler Hamiltonians, where the physics may be far easier to extract. The RG map therefore produces a dynamic map on Hamiltonians, and consecutive applications of this map generate a "flow" in the space of Hamiltonians. Often,

the form of the Hamiltonian is preserved, and the RG flow can be characterised by the trajectory of the parameters describing the Hamiltonian.

The development of RG methods has not only allowed sophisticated theoretical and numerical analysis of a broad range of many-body systems. It also explained phenomena such as *universality*, whereby many physical systems, apparently very different, exhibit the same macroscopic behaviour, even at a quantitative level. This is explained by the fact that these systems "flow" to the same fixed point under the RG dynamics.

For many condensed matter systems—even complex strongly interacting ones—the RG dynamics are relatively simple, exhibiting a finite number of fixed points to which the RG flow converges. Hamiltonians that converge to the same fixed point correspond to the same phase, so that the basins of attraction of the fixed points map out the phase diagram of the system. However, more complicated RG trajectories are also possible, including chaotic RG flows with highly complex structure[6–10]. Nonetheless, as with chaotic dynamics more generally, the structure and attractors of such chaotic RG flows can still be analysed, even if specific trajectories of the dynamics may be highly sensitive to the precise starting point. This structure elucidates much of the physics of the system[11–13]. RG techniques have become one of the most important technique in

[1]Department of Computer Science, University College London, London, UK. ✉e-mail: ucapjdj@ucl.ac.uk

modern physics for understanding the properties of complex many-body systems.

On the other hand, recent work has shown that determining the macroscopic properties of many-body systems, even given a complete underlying microscopic description, can be even more intractable than previously anticipated. In fact, refs. 14–16 showed that this goal is unobtainable in general: they engineered a quantum many-body Hamiltonian whose spectral gap, phase diagrams and any macroscopic property characterising a phase are uncomputable. These results imply that any RG technique which we may apply to this specific system in order to characterise the spectrum and other properties is bound to fail: there can be no RG scheme−or even more broadly, no algorithm−that can answer the spectral gap problem. Yet, it is unclear how such a negative result will emerge. In principle, this obstacle may be because there does not exist an RG map which can compute a coarse-grained version of an intractable Hamiltonian, or which cannot retain its macroscopic properties at every iteration, or again whose fixed points are not well-defined (or do not exist to begin with).

## Results

We denote a 2D $L \times L$ lattice as $\Lambda(L)$, and the minimum eigenvalue of a Hamiltonian $H$ (the ground-state energy) as $\lambda_0(H)$. After some RG procedure, we denote the renormalised Hamiltonian $R(H)$, and after $k$-iterations of the RG procedure $R^{(k)}(H)$. We also denote $\mathcal{B}(\mathcal{H})$ to be the set of bounded operators acting on Hilbert space $\mathcal{H}$.

The family of Hamiltonians we will consider is that from ref. 14, which are a set of translationally invariant, nearest neighbour, 2D spin-lattice models with open boundary conditions defined on $\Lambda(L)$. The Hamiltonians are parametrised by single parameter $\varphi$, and hence the set can be written as $\{H(\varphi)\}_{\varphi \in \mathbb{Q}}$. Each lattice site is associated with a spin system with local Hilbert space of dimension $d$, $\mathbb{C}^d$. The property of interest is the spectral gap, which is defined as the energy gap between the first excited state energy and the ground-state energy. Importantly, it is shown that as the lattice size goes to infinity, any Hamiltonian in this family must either have a spectral gap >1/2 or be gapless. However, determining which case occurs is undecidable.

Our main result is an explicit construction of a renormalisation group mapping for this Hamiltonian with the following features:

**Theorem 1 (Uncomputability of RG Flows (informal))** *We construct an RG map for the Hamiltonian of Cubitt, Pérez–García and Wolf*[14] *which has the following properties*:

1. The RG map is computable at each renormalisation step.
2. The RG map preserves whether the Hamiltonian is gapped or gapless, as well as other properties associated with the phase of the Hamiltonian.
3. The Hamiltonian is guaranteed to converge to one of two fixed points under the RG flow: one gapped, with low-energy properties similar to those of an Ising model with field; the other gapless, with low-energy properties similar to the critical XY-model.
4. The behaviour of the Hamiltonian under the RG mapping, and which fixed point it converges to, are uncomputable.

The undecidability of the fixed point follows implicitly from the undecidability of the spectral gap[14,15], since the fixed point depends on the gappedness of the unrenormalised Hamiltonian. Theorem 1 demonstrates that the renormalisation process fails, but not because it is impossible to construct a well-defined RG mapping: the actual reason is that the trajectory of the Hamiltonian under repeated applications of the RG mapping is itself uncomputable. Consequently, determining the fixed point that the trajectory eventually converges to is itself undecidable. This is despite each individual step of the RG process being computable.

We note a subtlety in the statement of Theorem 1. It is important that we are able to explicitly construct the RG scheme, rather than just prove the existence of such an RG scheme. If only existence were

proven, it would leave open the possibility that finding the RG scheme is itself an uncomputable task, thus meaning it cannot actually be determined.

## The Cubitt, Pérez–García and Wolf Hamiltonian

Before outlining our RG construction, we review some of the important features of the Hamiltonian from refs. 14, 15 used to prove the undecidability of the spectral gap. The Hamiltonian can be written as:

$$H(\varphi) = H_u(\varphi) \otimes \mathbb{1}_d \oplus 0 + \mathbb{1}_u \otimes H_d \oplus 0 + \mathbb{1}_{u,d} \oplus H_{trivial} + H_{guard}, \quad (1)$$

where $H_d \in \mathcal{B}(\mathcal{H}_d)$ is a Hamiltonian with a dense spectrum and zero ground-state energy, $H_{trivial} \in \mathcal{B}(\mathcal{H}_3)$ is a trivial Hamiltonian also with zero ground-state energy. $H_{guard} \in \mathcal{B}(\mathcal{H}_u \otimes \mathcal{H}_d \otimes \mathcal{H}_3)$ applies a large energetic penalty to states which have support on both $\mathcal{H}_u \otimes \mathcal{H}_d$ and $\mathcal{H}_3$. All of the undecidable physics is then contained in the part of the Hamiltonian $H_u(\varphi) \in \mathcal{B}(\mathcal{H}_u)$, and it is the ground-state energy of this Hamiltonian $H_u$ which determines whether the overall Hamiltonian is gapped or gapless.

## History states

To understand the structure of the $H_u(\varphi)$ ground state, we must first review how computation can be encoded in Hamiltonians and their ground states using *history states*. A quantum Turing Machine (QTM) is a model of quantum computation based on classical Turing Machines (TMs). Much like a classical Turing Machine, a QTM consists of a tape split up into cells, such that the cell is either empty or contains a symbol from an allowed set. The QTM also has a control *head* which moves along the tape. The head updates the tape at each time step depending on its internal state and the symbol currently written on the tape. The significant difference with respect to a classical TM is that the head and tape of a QTM can be in a superposition of states. The updates to the QTM and tape configuration are then described by a transition unitary, $U$, such that the overall state of the QTM updates as $|\psi\rangle \to U|\psi\rangle$ at each time step.

Given a particular QTM, using a construction of Gottesman and Irani[17], it is possible to encode the evolution of the QTM in the ground state of a specially constructed 1D nearest neighbour, translationally invariant Hamiltonian. In particular, the ground state is known as a *history state* and it encodes $T$ steps of the QTM computation. Here $T$ is a predefined and fixed function of the Hamiltonian's chain length determined by the particular QTM-to-Hamiltonian mapping. If the state of the QTM and its tape at time $t$ is $|\psi_t\rangle$, then the history state is

$$|\Psi_{hist}\rangle = \frac{1}{\sqrt{T}} \sum_{t=1}^{T} |t\rangle |\psi_t\rangle. \quad (2)$$

For the QTM-to-Hamiltonian mapping we are interested in, $T$ is an increasing function of the history state length, $T = T(L) = \Omega(2^L)$. Thus, longer-length history states encode more computational time steps.

## The ground state of $H_u(\varphi)$

The local Hilbert space which $H_u(\varphi)$ acts on can further be decomposed into a "classical" and "quantum" part: $\mathcal{H}_u = (\mathcal{H}_c)^{\otimes \Lambda(L)} \otimes (\mathcal{H}_q)^{\otimes \Lambda(L)}$. In particular, $H_u(\varphi)$ can be thought of as acting classically on states in $\mathcal{H}_c$. Furthermore, $H_u(\varphi)$ has the useful property that all its eigenstates are product states across these two parts of the Hilbert space. In particular, the ground state can be written as $|T\rangle_c \otimes |\psi_0\rangle_q$ where $|T\rangle_c \in (\mathcal{H}_c)^{\otimes \Lambda(L)}$ and $|\psi_0\rangle_q \in (\mathcal{H}_q)^{\otimes \Lambda(L)}$.

$H_u(\varphi)$ is designed so that $|T\rangle_c$ is the ground state of a classical Hamiltonian based on so-called Robinson tiles. That is, the local basis states in this part of the Hilbert space correspond to particular types of square tiles with markings on them, and the Hamiltonian enforces certain configurations of these tiles to be energetically penalised. Thus $|T\rangle_c$ corresponds to a non-penalised pattern of Robinson tiles. This

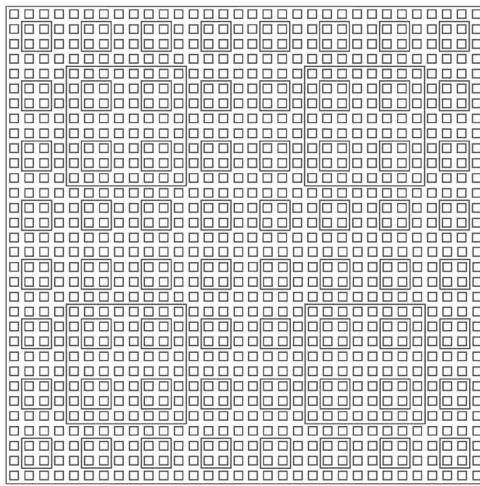

**Fig. 1 | Tiling pattern for the classical ground state $|T\rangle_c$.** Encoding the ground state of the classical Hamiltonian into Robinson tiles generates a quasi-periodic pattern of nested squares. History states are then placed along the top edge of every square in the pattern.

pattern has a self-similar structure of nested squares of increasing size, with side length $4^n + 1, n \in \mathbb{N}$[18] (see Fig. 1 for a diagram). $|\psi_0\rangle_q$ is coupled to $|T\rangle_c$ such that 1D history states (of the type described in Eq. (2)) appear along the top edge of every square in the pattern. Thus for every $n \in \mathbb{N}$, 1D history states of length $4^n + 1, n \in \mathbb{N}$, appear periodically across the lattice. Everywhere else in the lattice is in a trivial "filler" state which has zero energy.

The history states are designed to encode a QTM $M$ which takes input $\varphi \in \mathbb{Q}$ in binary (where $\varphi$ is the input parameter to the Hamiltonian), and then either halts or does not halt within the allotted time. By introducing an additional local penalty term to the Hamiltonian, the individual history states encoding a halting computation receive an energy penalty, and so the ground state of the whole lattice in the halting case picks up a positive energy contribution scaling as $\Omega(L^2)$. Conversely, in the non-halting case, the ground state of $H_u$ has energy going as $-\Omega(L)$. The overall ground state of the overall Hamiltonian $H(\varphi)$ is then either the zero energy ground state of $H_{trivial}$ in the halting case, or the ground state of $H_u(\varphi)$ in the non-halting case. In the halting case the ground-state energy scales as $\lambda_0(H_u(\varphi)) = \Omega(L^2)$, hence $H_u(\varphi)$ has a higher ground-state energy than $H_{trivial}$ and so the zero energy ground state of $H_{trivial}$ is the overall ground state. Otherwise, $\lambda_0(H_u(\varphi)) = -\Omega(L)$ and we see that the overall ground state is that of $H_u(\varphi)$. In the halting case, refs. 14, 15 show that $H(\varphi)$ is gapped, and in the non-halting case $H(\varphi)$ is gapless.

The key point for our purposes is that the overall behaviour of $H(\varphi)$ is determined by the ground-state energy of $H_u(\varphi)$. Since establishing whether a given universal Turing Machine halts is an undecidable problem[19], determining which ground state occurs, and thus whether the Hamiltonian is gapped or gapless, is undecidable.

## The block-spin renormalisation group (BRG)
Our RG map is based on a blocking technique widely used in the literature to study spin systems, often called the Block Spin Renormalisation Group (BRG)[20–23]. Note that this is also sometimes called the "quantum renormalisation group", but we will not use this name to avoid potential confusion. Modifications and variations of this RG scheme have also been extensively studied[24,25].

The BRG is among the simplest RG schemes. The procedure works by grouping nearby spins together in a block, and then determining the associated energy levels and eigenstates of this block by diagonalisation. Having done this, high-energy (or otherwise unwanted) states are removed, resulting in a new Hamiltonian.

As an explicit example, we repeat the review of the RG process in ref. 21 for the 1D isotropic XY-model defined below as:

$$H = -\frac{J}{2}\sum_{i=1}^{N-1}(X_i X_{i+1} + Y_i Y_{i+1}) + B\sum_{i=1}^{N} Z_i. \quad (3)$$

We first group terms into blocks of 2:

$$H = -\frac{J}{2}\sum_{\substack{i\,odd}}^{N-1}(X_i X_{i+1} + Y_i Y_{i+1}) - \frac{J}{2}\sum_{\substack{i\,even}}^{N-1}(X_i X_{i+1} + Y_i Y_{i+1}) + B\sum_{i=1}^{N} Z_i \quad (4)$$

$$= -\frac{J}{2}\sum_{\substack{i\,odd}}^{N-1}(X_i X_{i+1} + Y_i Y_{i+1}) + \sum_{\substack{i\,even}}^{N-1} h_i \quad (5)$$

where $h_i = -\frac{J}{2}(X_i X_{i+1} + Y_i Y_{i+1}) + BZ_i + BZ_{i+1}$ now contains all terms acting within the two site blocks. Diagonalising $h_i$ gives 4 states with energies $\{E_0^{(1)}, E_1^{(1)}, E_2^{(1)}, E_3^{(1)}\}$ in ascending order. We truncate the states associated with the two higher energies, and keep the lowest two which we label as $|0\rangle^{(1)}, |1\rangle^{(1)}$ with energies $E_0^{(1)}, E_1^{(1)}$, respectively. We now replace this operator with a new operator, acting on a single block-spin site with the form

$$\frac{(E_0^{(1)} - E_1^{(1)})}{2} Z_i^{(1)} + \frac{(E_0^{(1)} + E_1^{(1)})}{2} \mathbb{1}^{(1)}. \quad (6)$$

The between-block interaction now needs to be determined: to replicate this, we use $X = \xi^{(1)}X^{(1)}$, where $\xi^{(1)}$ can be determined by looking at the matrix elements under the new renormalised block basis, i.e., $\langle 0|^{(1)}X|1\rangle^{(1)} = \xi^{(1)}\langle 0|^{(1)}X^{(1)}|1\rangle^{(1)}$. The two new two-local terms acting on the block spins are then:

$$h_{i,i+1}^{(1)} = -\frac{J^{(1)}}{2}\sum_{\substack{i\,odd}}^{N/2-1}(X_i^{(1)} X_{i+1}^{(1)} + Y_i^{(1)} Y_{i+1}^{(1)}), \quad (7)$$

where $J^{(1)} = \xi^{(1)2}J$. By introducing an extra term depending on the identity, we find a renormalised Hamiltonian:

$$H^{(1)} = -\frac{J^{(1)}}{2}\sum_{\substack{i\,odd}}^{N/2-1}(X_i^{(1)} X_{i+1}^{(1)} + Y_i^{(1)} Y_{i+1}^{(1)}) + B^{(1)}\sum_{i=1}^{N/2} Z_i^{(1)} + C^{(1)}\sum_{i=1}^{N/2} \mathbb{1}_i^{(1)}, \quad (8)$$

where $C^{(1)} = (E_0^{(1)} + E_1^{(1)})/2$. After $n$ iterations of the RG mapping, we have a Hamiltonian

$$H^{(n)} = -\frac{J^{(n)}}{2}\sum_{\substack{i\,odd}}^{N/2^n-1}(X_i^{(n)} X_{i+1}^{(n)} + Y_i^{(n)} Y_{i+1}^{(n)}) + B^{(n)}\sum_{i=1}^{N/2^n} Z_i^{(n)} + C^{(n)}\sum_{i=1}^{N/2^n} \mathbb{1}_i^{(n)}, \quad (9)$$

where the constants are defined by the same procedure: $J^{(n)} = \xi^{(n)2}J^{(n-1)}$, $B^{(n)} = B^{(n-1)} + (E_0^{(n)} - E_1^{(n)})/2$, $C^{(n)} = C^{(n-1)} + (E_0^{(n)} + E_1^{(n)})/2$.

## Our RG scheme
We now want to construct an RG scheme for $H(\varphi)$ which preserves the relevant physical properties. Most notably, whether the Hamiltonian is gapped or gapless. We will show that in order to preserve the low-energy properties of this Hamiltonian, we can reduce the analysis to finding RG schemes for each of the Hamiltonians in Eq. (1). Well-defined RG schemes exist for $H_d, H_{trivial}, H_{guard}$ which preserve their gaps and ground-state energies, hence the remaining task is finding an RG scheme for $H_u(\varphi)$. In particular, we develop an RG scheme which

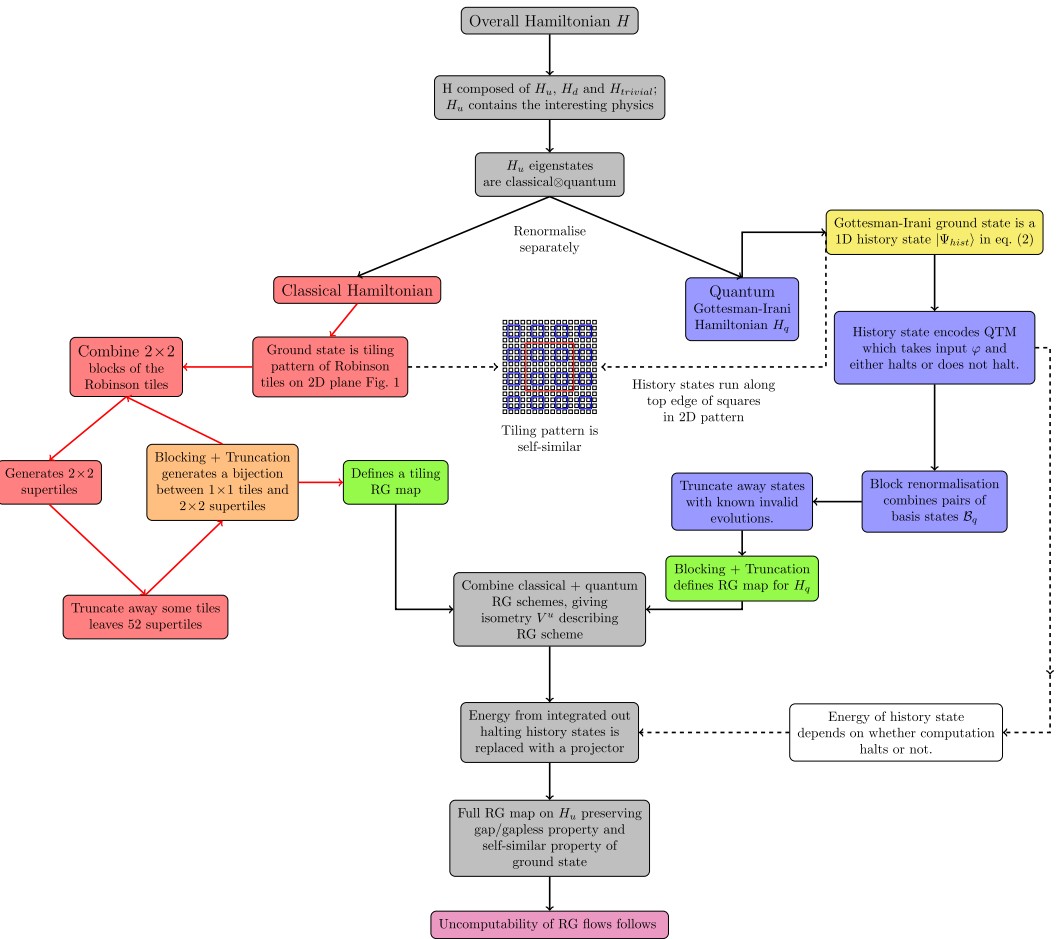

**Fig. 2 | Flow diagram of the proof.** The idea behind our proof construction is to separate $H_u$ into a classical and a quantum part. The RG scheme for the classical part is described on the left-hand side of the flow chart, and the right-hand side shows the RG scheme for the quantum part. The separate classical and quantum schemes are then combined (see the grey boxes) to create an RG scheme for the entire Hamiltonian.

maps the following:

$$
\begin{aligned}
\mathcal{R} : H_u(\varphi) \otimes \mathbb{1}_d \oplus 0 + \mathbb{1}_u \otimes H_d \oplus 0 + \mathbb{1}_{u,d} \oplus H_{trivial} + H_{guard} \\
\rightarrow R(H_u(\varphi)) \otimes R(\mathbb{1})_d \oplus 0 + R(\mathbb{1})_u \otimes R(H_d) \oplus 0 \\
+ R(\mathbb{1})_{u,d} \oplus R(H_{trivial}) + R(H_{guard}),
\end{aligned}
\tag{10}
$$

allowing us to break the problem of finding an overall RG scheme into finding one for each individual Hamiltonian.

To retain the properties of the overall Hamiltonian, the RG scheme must maintain the ground-state energy density of $H_u(\varphi)$ in both the halting and non-halting cases. We will do this by: (a) preserving the overall self-similar structure of the Robinson tiling and thus the pattern of the history states appearing in the ground state, (b) ensuring that the energy contribution of each individual history states is preserved. Since the history states give the only non-trivial energy contribution to the ground-state energy of $H_u(\varphi)$, then this is sufficient for our purposes.

**The RG scheme for $H_u(\varphi)$**
In order to develop an RG scheme for $H_u(\varphi)$, we remark that its eigenstates are product states across $(\mathcal{H}_c)^{\otimes \Lambda(L)} \otimes (\mathcal{H}_q)^{\otimes \Lambda(L)}$. This allows us to split our RG scheme up further into one part that renormalises the classical space $(\mathcal{H}_c)^{\otimes \Lambda(L)}$ and another for the quantum space $(\mathcal{H}_q)^{\otimes \Lambda(L)}$ (a rigorous justification of this is given in Section E of the Supplementary Information). As with the BRG, both schemes consist of a blocking and truncation procedure. We give a flow diagram of the proof in Fig. 2.

**The blocking procedure.** The RG scheme proceeds by splitting the lattice into disjoint $2 \times 2$ square blocks. The basis states of the individual lattice sites within a $2 \times 2$ block are then combined into a single site on a new lattice, such that if the initial local Hilbert space dimension was $d$, then the new lattice sites have local Hilbert space dimension $d^4$. Having obtained a new reduced lattice of size $L/2 \times L/2$, we now wish to reduce the size of the local basis to only include basis states which contribute to low-energy states.

**Truncating the classical space.** Since the basis states in the classical part of the Hilbert space are represented by Robinson tiles, the new renormalised-basis states correspond to all possible combinations of these tiles on a $2 \times 2$ block: we call these "supertiles". However, a subset of these bigger tiles can be shown to either have high energy with respect to the previous Hamiltonian or will be removed at later stages of the RG process. Thus removing these supertiles will only remove local basis states which do not contribute to the low-energy states of the renormalised Hamiltonian. It turns out that each new state in the renormalised basis can be identified in a one-to-one manner with a state in the unrenormalised basis, such that the Hamiltonian is of the same form. Thus, the ground state of the new renormalised Hamiltonian on the classical part of the Hilbert space will not only be self-similar but will generate the same Robinson pattern as the unrenormalised ground state. A detailed analysis of the RG scheme for this part of the Hamiltonian is given in Section C of the Supplementary Information.

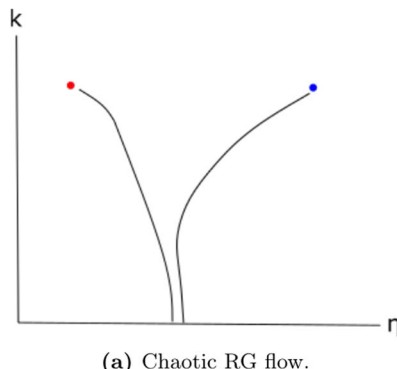

**(a)** Chaotic RG flow.

**(b)** Uncomputable RG flow.

**Fig. 3 | Chaotic vs Uncomputable RG flow behaviour.** In both diagrams, $k$ represents the number of RG iterations and $\eta$ represents some parameter characterising the Hamiltonian; the blue and red dots are fixed points corresponding to different phases. We see that in the chaotic case (**a**), the Hamiltonians diverge exponentially in $k$, according to some Lyapunov exponent. In the undecidable case (**b**), the Hamiltonians remain arbitrarily close for some uncomputably large number of iterations, whereupon they suddenly diverge to different fixed points.

**Truncating the quantum space.** Finally, we consider the effect of the blocking procedure on the history state, combining pairs of cells on the Turing Machine tape. After $k$-iterations of the blocking, a single new basis state will contain $2^k$ Turing Machine tape cells on a single lattice site, where at each iteration, it is possible to further remove some of the states which we know must have high energy. For example, there exist sets of states which are a priori known to be energetically penalised, e.g., states corresponding to Turing Machine configurations with two heads next to each other. Such states are known to not contribute to the ground state, thus they can be removed from the local Hilbert space in the truncation procedure. We are also able to discard some states which are guaranteed to evolve into one of these disallowed states. Furthermore, after iterating the RG procedure multiple times, there will be entire history states localised to a single renormalised-basis state. We can then integrate these out. More details are given in Section D of the Supplementary Information.

**Energy contribution of the integrated out states**
We have glossed over some details in the previous section. In particular, what happens to the energy contributions from the history states which are integrated out?

When the RG mapping has been applied $k$ times, such that $2^k \geq 4^n + 1$, then we know that a full history state which would appear in the ground state of $H(\varphi)$ is now formed from a superposition of basis states on a single site. By diagonalising the on-site Hamiltonian, we see that it now forms the lowest energy state of the local Hilbert space. As discussed earlier, in the halting case, this history state will pick up some positive energy, which is known explicitly as per ref. 26, and in the non-halting case it has exactly zero energy. In order to preserve the ground-state energy of the overall Hamiltonian, when integrating out the local basis states, we take the energy contribution of the history state and add it to a local projector term. This has the effect of introducing a local energy shift which preserves the overall energy. This is equivalent to introducing the term $C^{(n)}\mathbb{1}_i$ in the BRG procedure as per Eq. (9). See Supplementary Information E.1 and E.2 for more details.

This introduces a 1-local term in $H_u(\varphi)$ which has the form $\tau_2(k)\mathbb{1}_i$ acting on each lattice site $i$ where, if the encoded TM is non-halting on input $\varphi$, then $\tau_2(k) = -2^{-k} \forall k$. If the TM halts on input $\varphi$ then:

$$\tau_2(k) = \begin{cases} -2^k & k < k_h(\varphi) \\ -2^k + \Omega(4^{k-k_h(\varphi)}) & k \geq k_h(\varphi). \end{cases} \quad (11)$$

Here, $k_h(\varphi)$ is defined as the following: let $L_h \in \{4^n+1\}_{n\in\mathbb{N}}$ be the smallest-length history state for which the TM $M$ halts when running on input $\varphi$, then $k_h$ is the smallest integer satisfying $2^{k_h(\varphi)} > L_h(\varphi)$. The behaviour of $\tau_2(k)$ is fully discussed in Supplementary Information E.7.

We see that after $k$-iterations, the Hamiltonian $R^{(k)}(H_u(\varphi))$ has ground-state energy which scales as

$$\lambda_0(R^{(k)}(H_u(\varphi))) = \begin{cases} +\Omega(L^2) & \text{if } M \text{ halts on } \varphi \\ -\Omega(L) & \text{if } M \text{ is non-halting on } \varphi. \end{cases} \quad (12)$$

A crucial feature is that every step of the RG process is explicitly computable: it is simply a case of blocking together four sites, determining the renormalised-basis states, and removing subsets of local basis states which do not contribute to the low-energy subspace. Even determining whether a given history state contains a halting computation or not can be done by examining the legitimate evolution encoded within the history state, finding whether that halts or not, and then integrating out its energy contribution appropriately. The time taken is a function of the number of local basis states on each site which is upper bounded by $O(d^{4k})$. Thus each step of the RG procedure is computable, as claimed in point 1 of Theorem 1.

**The RG trajectory**
As per Eq. (11), we see that the Hamiltonian has a coefficient which is exactly $-2^k$ in the case the encoded TM does not halt. However, in the halting case, $\tau_2(k)$ begins to change behaviour as soon as the number of spins that have been blocked together is larger than the length of the history state needed to encode a halting computation. Thus, the $k$ for which $\tau_2(k)$ changes behaviour depends on the length when the Turing Machine first halts, and hence on the time step at which the Turing Machine halts. However, as we pointed out before, this quantity is undecidable in general, and thus determining whether $\tau_2(k)$ eventually becomes positive is itself uncomputable.

Furthermore, there are two fixed points associated with the RG flow. One occurs for $\tau(k) = -2^k$ which corresponds to a gapless Hamiltonian, and the other for $\tau(k) \to -2^k + \Omega(4^{k-k_h(\varphi)})$, which corresponds to a gapped Hamiltonian. Since distinguishing between these two cases is undecidable, our argument immediately yields that:

**Corollary 2** *Determining whether the Hamiltonian flows to the gapped or gapless fixed point under this RG scheme is undecidable.*

Indeed, the Hamiltonian from ref. 14 has two very different fixed points: one which at low energies roughly corresponds to a 2D Ising model and another which corresponds to a critical, gapless XY-model (further discussion in Section G of the Supplementary Information).

Thus, we have constructed an RG scheme for the Hamiltonian $H(\varphi)$ which is computable at every step, but the overall trajectory and end-point is uncomputable.

## Discussion

In this work, we have shown that a qualitatively new type of RG flow occurs in many-body Hamiltonians with undecidable spectral gap. Specifically, we give an explicit construction and analysis a block-spin RG procedure for the Hamiltonian of ref. 14 which we are able to study analytically and prove that it has the following features: (i) the RG map is computable at each renormalisation step; (ii) the RG map preserves whether the Hamiltonian is gapped or gapless; (iii) the Hamiltonian is guaranteed to converge to one of two fixed points under the RG flow; (iv) the behaviour of the Hamiltonian under the RG mapping, the trajectory of the RG flow and which fixed point it converges to are all uncomputable.

We show that under this RG construction, the Hamiltonian flows toward one of two RG fixed points: either a gapped Ising-like Hamiltonian or a gapless critical XY-like Hamiltonian. Furthermore, the parameters characterising the Hamiltonian have a trajectory depending on the halting time of the Turing Machine encoded within the Hamiltonian. Since the Halting Problem is undecidable and the halting time uncomputable, the trajectory of the Hamiltonian under the RG flow−and therefore which fixed point it ultimately converges to−are uncomputable, even if the parameters of the initial Hamiltonian are known exactly.

This is a qualitatively new and more extreme form of unpredictability that goes beyond even chaotic RG flows which have been previously studied. The unpredictability of chaotic systems arises from the fact that even a tiny difference in the initial system parameters−which in practice may not known exactly−can eventually lead to exponentially diverging trajectories (see Fig. 3). However, the more precisely the initial parameters are known, the longer it is possible to accurately predict the trajectory of a chaotic process, and if the system parameters were known exactly, then in principle it becomes possible to determine the long-time behaviour of the RG flow. The RG flow behaviour exhibited in this work is more intractable still. Even if we know the *exact* initial values of all system parameters, its RG trajectory and the fixed point it ultimately ends up at is provably impossible to predict. Moreover, no matter how close two sets of initial parameters are, it is impossible to predict how long their trajectories will remain close together before abruptly diverging to different fixed points that correspond to separate phases (see Fig. 3). Thus, the structure of the RG flow− e.g., the basins of attraction of the fixed points−is so complex that it cannot be computed or approximated, even in principle. We note that a similar form of unpredictability has previously been seen in classical single-particle dynamics, in seminal work by Moore[27–29], while our result shows for the first time that this extreme form of unpredictability can occur in RG flows of many-body systems.

Despite the somewhat artificial Hamiltonian considered here, we expect the behaviour of the RG scheme here to be generic, in the following sense. For any well-defined, computable RG scheme for Hamiltonians with undecidable macroscopic properties, we expect that at least one coefficient of a relevant operator should have an uncomputable trajectory. The reasoning is straightforward: the well-definedness and computability of the RG flow implies that, at each step of the RG process, we would be able to find each parameter characterising the Hamiltonian after each iteration. However, when the macroscopic properties of the Hamiltonian are undecidable, we expect determining which fixed point it flows towards to be an undecidable problem. For there to be no contradiction between these two statements, the parameters of the Hamiltonian must flow in an uncomputable manner (otherwise, the entire flow is computable and we reach a contradiction). As such, the uncomputable behaviour observed in the RG scheme here must occur *for any RG scheme* one can construct for Hamiltonians whose macroscopic properties are uncomputable from its microscopic description (note that ref. 16 has shown that such Hamiltonians can constitute a non-zero-measure

subset of a phase diagram, so do not require arbitrarily precisely tuned parameters).

Often RG flows are characterised by a set of continuous differential equations. By the nature of having a discretised lattice and a real space RG procedure, it is not natural to consider continuous variation of the parameters in terms of differential equations[30]. Rather, the RG relations in this setting are expressed in terms of finite difference equations, e.g., for a Hamiltonian characterised by a set of parameters $\{\alpha_i\}_i$, such that after the $k^{th}$ RG iteration the coefficients are denoted $\{\alpha_i(k)\}_i$, then:

$$\alpha_i(k) - \alpha_i(k-1) = f_i(k, \{\alpha_j(k-1)\}_j). \qquad (13)$$

In the case of the uncomputable RG flows exhibited here, $f_i(k, \{\alpha_j\}_j)$ will be some function whose behaviour is uncomputable as we iterate $k$ and the coefficients $\{\alpha_j\}_j$. In the case of $\tau_2(k)$ for the block-spin RG scheme, we have constructed in this work, $f$ depends on whether a given TM halts after a time depending on $k$. For RG flows characterised by continuous differential equations, we expect there should exist RG schemes with uncomputable behaviour that satisfy analogous differential equations: $\partial\alpha_i/\partial k = f_i(k, \{\alpha_j(k-1)\}_j)$, where $f$ is again an uncomputable function. In the continuous case, one would expect similar behaviour to that observed here: a particular parameter travels along a well-defined trajectory, but at some uncomputable point abruptly changes its behaviour and diverges from its previous trajectory.

Naturally, there are limitations on the generality of the conclusions that can be drawn from this work in the sense that the Hamiltonian discussed in this work is highly artificial and the RG scheme reflects this. Indeed, this Hamiltonian has an enormous local Hilbert space dimension and its matrix elements are highly artificially tuned. Both of these factors are unlikely to be present in naturally occurring Hamiltonians. A step towards overcoming this limitation was taken in ref. 16, where it was shown that Hamiltonians with uncomputable properties can occupy a non-zero-measure set of the phase diagram, thus do not depend on arbitrarily precise parameter tuning. As the Hamiltonians in that work are a development of the Cubitt–Pérez-García–Wolf Hamiltonian we have studied here, we expect our results will can readily be extended to this case (and indeed to the Hamiltonian in ref. 31 which also displays undecidable properties). However, the Hamiltonians remain highly artificial. Thus an obvious route for further work is to look for more natural Hamiltonians displaying undecidable behaviour and consider RG schemes to renormalise them.

Furthermore, although the RG scheme is essentially a simple BRG scheme, the details of our construction and analysis rely on knowledge of the structure of the ground states. Due to the behaviour of this undecidable model, any BRG scheme will have to exhibit similar behaviour to the one we have analysed rigorously here. But it would be of interest to find a simpler RG scheme for this Hamiltonian (or other Hamiltonians with undecidable properties) which is able to truncate the local Hilbert space to a greater degree, without using explicit a priori knowledge of the ground state, whose behaviour can still be analysed rigorously.

It is also worth noting that the Hamiltonian and RG schemes constructed here could also be used to prove rigorous results for chaotic (but still computable) RG flows. Indeed, if we modify the Hamiltonian $H(\varphi)$ so that instead of running a universal Turing Machine on input $\varphi$, it carries out a computation of a (classical) chaotic process (e.g., repeated application of the logistical map), then two inputs which are initially very close may diverge to completely different outputs after some time. By penalising this output qubit appropriately, the Hamiltonian will still flow to either the gapped or gapless fixed point depending on the outcome of the chaotic process under

our RG map, but the RG flow will exhibit chaotic rather than uncomputable dynamics.

## Data availability
Data sharing not applicable to this article as no datasets were generated or analysed during this study.

## Code availability
An accompanying *Mathematica* notebook is available at https://gitlab.com/ucl_cs_quantum/RG_flows_uncomputability.

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

## Acknowledgements
E.O. and T.S.C. are supported by the Royal Society. J.D.W. is supported by the EPSRC Centre for Doctoral Training in Delivering Quantum Technologies (grant EP/L015242/1). This work has been supported in part by the EPSRC Prosperity Partnership in Quantum Software for Simulation and Modelling (grant EP/S005021/1), and by the UK Hub in Quantum Computing and Simulation, part of the UK National Quantum Technologies Programme with funding from UKRI EPSRC (grant EP/T001062/1).

## Author contributions
J.D.W., E.O. and T.S.C. all contributed significantly to the ideas, technical approaches and proofs in this work. All authors contributed to drafting, revising and writing the paper.

## Competing interests
The authors declare no competing interests.
