## [Peer Review File · Nature Communications]

REVIEWER COMMENTS

Reviewer #1 (Remarks to the Author):

```
\documentclass{letter}
```

```
\usepackage{graphicx}
```

```
\usepackage{epsfig}
```

```
\usepackage{multicol}
```

```
\usepackage{amsmath}
```

```
\usepackage{amssymb}
```

```
\usepackage{bbm}
```

```
\begin{document}
```

This is the continuation of a series of works by Toby S. Cubitt and various co-authors on the undecidability of the spectral gap in quantum systems. In particular, it is related to papers [7] and [8] and examines the behavior of renormalization group (RG) flows in the model examined in these references. The Hamilton operator constructed in [7] contains an external parameter and is connected to a universal Turing machine (UTM). The Hamilton operator has a gap when the UTM stops on an input related to the parameter. Consequently, the spectral gap problem of the Hamilton operator is equivalent to the stopping problem of the UTM, i.e. undecidable.

In the present work, the authors construct an RG treatment that renormalizes such a Hamilton operator whose Hilbert space is decomposed as follows:

```
%
```

```
\begin{equation}
```

```
\{\mathcal{H}\} = |0\rangle \oplus \{\mathcal{H}\}_u \otimes \{\mathcal{H}\}_d;
```

```
\end{equation}
```

```
%
```

with $|0\rangle$: zero-energy filler state, $\{\mathcal{H}\}_u$: Hilbert space of the Hamiltonian with undecidable ground state and $\{\mathcal{H}\}_d$: Hilbert space of the Hamiltonian having a dense spectrum. Furthermore $\{\mathcal{H}\}_u$ is composed of a classical "tiling layer" and a highly entangled "quantum layer". The steps of renormalisation of \mathcal{H}_u follows the method in [7] and the RG map of the complete Hamiltonian is computable and in principle ends at two possible fixed points, yet the resulting RG flow is uncomputable.

The topic of the work is interesting in the field of quantum systems, the results obtained are valuable, even if not completely unexpected in the light of the previous results in [7,8]. When constructing the RG procedure explicitly, some points need to be clarified.

i) In Definition 4. the mapping is made from one set of r -local interactions, with $d \in \mathbb{N}$ to a new set of r' -local interactions, with $d' \in \mathbb{Z}$. Here $r' \leq r$ is required, which is generally not fulfilled in higher dimensions. Also during the RG process, the increase in parameters can lead to $d' > d$.

ii) In definition 4, point 3, the transformation properties of the order parameter are defined. Before doing this, however, some properties of the order parameter should be explained. Is it a scalar, a vector, or can it be a function? This should be done before the explicit definition in Section 2.5.

iii) The Block Spin Renormalisation Group (BRG) is introduced in Section 2.2.1. The truncation process is explained, but nothing is said about the way in which the renormalised parameters are calculated. Another point where the upper limits of the summations in (2.9) appear wrong: $N/2$ should be replaced with $N/2^n$.

iv) In Section 2.2.2 the magnetisation of the 1D Ising model should be $m = \frac{1}{N} \sum_{i=1}^N \sigma_i$.

v) In Eq.(2.22) the expression $\theta(1/T^2)$ should be defined.

vi) Another important point: all over the paper H_d is identified as the critical XY-model. In the supplementary information it is defined with the local term: $X_i \otimes X_{i+1} + Y_i \otimes Y_{i+1} + Z_i \otimes \mathbb{1}^{(i+1)} + \mathbb{1}^{(i)} \otimes Z_{i+1}$. The spectrum of this Hamiltonian, however is gapped. See in Ref.[15], or in E. Barouch and B. McCoy, Phys. Rev. A 3, 786 (1971). The model is critical if the transverse field term, $h(Z_i \otimes \mathbb{1}^{(i+1)} + \mathbb{1}^{(i)} \otimes Z_{i+1})$, has a strength $0 \leq h \leq 1/2$. This critical model was renormalized by the BRG in [15] and, depending on the block length, several fixed points were identified. This result should be mentioned and discussed.

`\end{document}`

Reviewer #2 (Remarks to the Author):

The authors construct a renormalisation group (RG) flow for a specific model for which it is uncomputable in which phase the flow drives a particular initial state. They contrast this behaviour with that of a chaotic flow, which is still computable but where small uncertainties in the input parameters can turn into large differences after renormalisation.

Before I present my viewpoint on the manuscript, I should say that I do not think that I am the target audience for this paper. While I do have a background in renormalisation, my focus is more on the computation and less on mathematical rigour. In this sense, I am evaluating the manuscript from some distance.

Having said that, I think that the existence of such systems and RG flows is definitely interesting, and could have an important impact on real world systems. However, I also think that the manuscript leaves some important questions open, and its style/readability can be improved.

More concretely, my main question (which is partially and very briefly raised by the authors themselves in the conclusions) is how relevant this mathematical example is for actual physical systems. Is this observed behaviour generic in any way? Is it a “bug” of the specific RG map, or a genuine feature of the physical system? If it is a feature of the system, how would one see the uncomputability e.g. in the standard continuum RG, where the RG flow is specified in terms of differential equations? While I understand that it might be difficult to make such statements without sacrificing mathematical rigour, there should at least be some more discussion. This would also connect the work to the wider literature.

Regarding style, I find the manuscript pretty difficult to read. This is mainly due to a mixture of countless typos/word omissions/missing definitions in the main text, which gives the paper a careless appearance. To name a few explicit points:

- References [7] and [8] seem to be literally the same, and the authors often cite both in the same place. This should not happen, in particular because one of the present authors is also an author of the referenced paper. Also, the references should be sorted in some way (right now the first reference in the text is [30]).

- Many of the theorems and definitions are not phrased as such. For example, Theorem 1 says that the authors construct an RG map with some properties, while the statement probably should be that such an RG map exists. While this seems like a minor point, it leaves it to the reader to guess what the precise statement is, which in my opinion should be avoided in a mathematical (or in fact any) paper.

- In eq. (2.9), the upper limit of the summations should probably be $N/(2^n)$ since every RG step halves the number of lattice sites.

- In the main result, a family F is introduced, but nothing more is said about it. Clearly there is more information in the supplemental material, but I think that the key result, including necessary definitions, should be contained in the main paper.

Beyond these stylistic points, in my view the paper would be more easily accessible if more simple examples would be given on the way to clarify the important concepts. For example, it would be helpful to give concrete local interactions h^{row} and h^{col} for e.g. a 2×2 lattice. Another suggestion would be to give a brief overview of the key steps of the proof in terms of a graph/map to have a better visualisation of what is going on.

Response to Referees: Uncomputably Complex Renormalisation Group Flows

James D. Watson,^{*} Emilio Onorati,[†] and Toby S. Cubitt[‡]

Department of Computer Science, University College London, UK

We sincerely thank the Referees for their very valuable reports and for their positive assessment regarding the importance and impact of our results concerning RG flows. We have also appreciated their insightful comments and criticisms, which we have carefully addressed when revising our manuscript. We provide a point-by-point answer after the following summary.

Summary of changes: As well as addressing the individual reviewer's comments, we have completely rewritten the main text of the paper to a clearer exposition of the results and proof to a general audience. The more detailed overview which previously formed the main text, as well as the proof details, are now contained at the start of the supplementary information. Corrections suggested by the reviewers to what was previously the main text have now been incorporated into the relevant sections in the supplementary material. We note that we have also heavily rewritten the Preliminaries and Proof Overview sections of the Supplementary Information. Some of the specific changes requested have been highlighted in red text.

Reviewer 1

✓ i) In Definition 4. the mapping is made from one set of r -local interactions, with $d \in \mathbb{N}$ to a new set of r' -local interactions, with $d' \in \mathbb{Z}$. Here $r' \leq r$ is required, which is generally not fulfilled in higher dimensions. Also during the

^{*}ucapjdj@ucl.ac.uk

[†]e.onorati@ucl.ac.uk

[‡]t.cubitt@ucl.ac.uk

RG process, the increase in parameters can lead to $d' > d$.

The statement $r' > r$ has been removed. Both d', r' are now simply stated to be natural numbers.

✓ ii) In definition 4, point 3, the transformation properties of the order parameter are defined. Before doing this, however, some properties of the order parameter should be explained. Is it a scalar, a vector, or can it be a function? This should be done before the explicit definition in Section 2.5.

A paragraph has been added to the “Preliminaries” subsection “Notation” which defines an order parameter.

✓ iii) The Block Spin Renormalisation Group (BRG) is introduced in Section 2.2.1. The truncation process is explained, but nothing is said about the way in which the renormalised parameters are calculated. Another point where the upper limits of the summations in (2.9) appear wrong: $N/2$ should be replaced with $N/2^n$.

A more explicit example of the BRG method is now given and how the constants are calculated has now been included. The limits on the appropriate sum have been corrected.

✓ iv) In Section 2.2.2 the magnetisation of the 1D Ising model should be $m = \frac{1}{N} \sum_{i=1}^N \sigma_i$.

This has been changed appropriately.

✓ v) In Eq.(2.22) the expression $\theta(1/T^2)$ should be defined.

We apologise – theta θ should have been capital theta Θ for the asymptotic notation rather than lower case.

✓ vi) Another important point: all over the paper H_d is identified as the critical XY-model. In the supplementary information it is defined with the local term: $X_i \otimes X_{i+1} + Y_i \otimes Y_{i+1} + Z_i \otimes \mathbb{1}^{(i+1)} + \mathbb{1}^{(i)} \otimes Z_{i+1}$. The spectrum of this Hamiltonian, however is gapped. See in Ref.[15], or in E. Barouch and B. McCoy, Phys, Rev. A 3, 786 (1971). The model is critical if the transverse field term, $h(Z_i \otimes \mathbb{1}^{(i+1)} + \mathbb{1}^{(i)} \otimes Z_{i+1})$, has a strength $0 \leq h \leq 1/2$. This critical model was renormalized by the BRG in [15] and, depending on the block length, several fixed points were identified. This result should be mentioned

and discussed.

We have rewritten the section in the Supplementary Information concerning the RG procedure for H_d so that it now makes reference to [15] and discusses appropriate parameters and scaling.

Reviewer 2

✓ (1) I also think that the manuscript leaves some important questions open, and its style/readability can be improved.

We have completely rewritten the main text of the article and hope that it addresses this criticism. In particular, we have drastically reduced the amount of detail we go into, while hopefully giving an idea of how to proof works. Furthermore, the “Overview of the Proof” section has now been moved to Supplementary Information and had been rewritten in a way which is hopefully easier to read.

✓ (2) More concretely, my main question (which is partially and very briefly raised by the authors themselves in the conclusions) is how relevant this mathematical example is for actual physical systems. Is this observed behaviour generic in any way? Is it a “bug” of the specific RG map, or a genuine feature of the physical system? If it is a feature of the system, how would one see the uncomputability e.g. in the standard continuum RG, where the RG flow is specified in terms of differential equations? While I understand that it might be difficult to make such statements without sacrificing mathematical rigour, there should at least be some more discussion. This would also connect the work to the wider literature.

In the main text, in the “Conclusions and Discussions” section, we have now added a paragraphs titled “How generic is this behaviour?” and “Comparison with Continuous RG Schemes” in which we address these questions.

✓ (3) Regarding style, I find the manuscript pretty difficult to read. This is mainly due to a mixture of countless typos/word omissions/missing definitions in the main text, which gives the paper a careless appearance.

As noted above, we have completely rewritten the main text, and have clarified first two sections of the Supplementary Information. We hope that the revised text is more accessible.

✓ (4) *References [7] and [8] seem to be literally the same, and the authors often cite both in the same place. This should not happen, in particular because one of the present authors is also an author of the referenced paper. Also, the references should be sorted in some way (right now the first reference in the text is [30]).*

These are two separate papers, describing the same result but published in two separate journals and containing very different content. The Nature version presents the results addressed at a general science audience, but does not contain the rigorous proofs of these results. The long 100+ page paper now published in Forum of Mathematics Pi contains the full, rigorous, mathematical proofs. We believe that citing both is relevant, as the long technical mathematics paper contains relevant technical details, while the shorter Nature paper is significantly more accessible to a general science audience.

✓ (5) *Many of the theorems and definitions are not phrased as such. For example, Theorem 1 says that the authors construct an RG map with some properties, while the statement probably should be that such an RG map exists. While this seems like a minor point, it leaves it to the reader to guess what the precise statement is, which in my opinion should be avoided in a mathematical (or in fact any) paper.*

There is a critical difference between merely proving existence and giving an explicit construction, which we failed to make sufficiently clear. This is a particularly important subtlety in the context of uncomputability. The existence of something does not necessarily imply that it is computable¹. Stating that our RG scheme “exists” without showing that it can be “explicitly constructed” or “efficiently computed” leaves open the possibility that the definition of RG scheme is itself uncomputable, and that this is responsible for the uncomputable behaviour of the Hamiltonian under the RG procedure (which would be a trivial result). Hence noting that the RG scheme is constructable is crucial and necessary.

With this in mind, we have tried to clarify this subtle but important distinction, by including a paragraph explaining why this statement is im-

¹A trivial example of this might be Chaitin’s constant, which is a well defined real number, but which is uncomputable.

portant to the theorem statement beneath Theorem 1 in the main text. We thank the reviewer for drawing this to our attention.

✓ (6) *In eq. (2.9), the upper limit of the summations should probably be $N/(2^n)$ since every RG step halves the number of lattice sites.*

We agree — this has been changed appropriately.

(7) *In the main result, a family F is introduced, but nothing more is said about it. Clearly there is more information in the supplemental material, but I think that the key result, including necessary definitions, should be contained in the main paper.*

Any reference to F in the main text has now been removed.

(8) *Beyond these stylistic points, in my view the paper would be more easily accessible if more simple examples would be given on the way to clarify the important concepts. For example, it would be helpful to give concrete local interactions h^{row} and h^{col} for e.g. a 2×2 lattice.*

Unfortunately, for the particular Hamiltonian we are working with in the paper, the local Hilbert space dimension of the Hamiltonian we consider is of the order $\sim 10^6$, which makes writing down the interactions concretely unfeasible even for a 2×2 lattice. We sympathise with the reviewer's point here! The interactions are specified in great detail in Cubitt, Perez-Garcia and Wolf, but since the interactions encode the transition rules of a Quantum Turing Machine they are highly complex and writing them out explicitly (even partially) is unlikely to be useful to readers.

We have attempted to partially address this by writing down an explicit example of the analogous construction for the much simpler XY model, before giving the construction for the far more complicated Hamiltonian we are concerned with in our results.

(9) *Another suggestion would be to give a brief overview of the key steps of the proof in terms of a graph/map to have a better visualisation of what is going on.*

Thank you for this suggestion — we have now added a flow-chart of the proof structure to the main text, and a more detailed version of the flow chart to the supplementary information.

REVIEWERS' COMMENTS

Reviewer #1 (Remarks to the Author):

The authors have successfully addressed the issues I raised in my report. I propose that the manuscript be published as is.